# Undefeated—Changing the phenamacril scaffold is not enough to beat resistant *Fusarium*

**Rasmus D. Wollenberg**[1], **Søren S. Donau**[1], **Manuel H. Taft**[2], **Zoltan Balázs**[1], **Sven Giese**[2], **Claudia Thiel**[2], **Jens L. Sørensen**[3], **Thorbjørn T. Nielsen**[1], **Henriette Giese**[1], **Dietmar J. Manstein**[2], **Reinhard Wimmer**[1], **Teis E. Sondergaard**[1]*

**1** Department of Chemistry and Bioscience, Aalborg University, Aalborg, Denmark, **2** Institute for Biophysical Chemistry, OE4350, Hannover Medical School, Hannover, Germany, **3** Department of Chemistry and Bioscience, Aalborg University, Esbjerg, Denmark

* tes@bio.aau.dk

**Data Availability Statement:** All relevant data are within the manuscript and its Supporting Information files.

## Abstract

Filamentous fungi belonging to the genus *Fusarium* are notorious plant-pathogens that infect, damage and contaminate a wide variety of important crops. Phenamacril is the first member of a novel class of single-site acting cyanoacrylate fungicides which has proven highly effective against important members of the genus *Fusarium*. However, the recent emergence of field-resistant strains exhibiting qualitative resistance poses a major obstacle for the continued use of phenamacril. In this study, we synthesized novel cyanoacrylate compounds based on the phenamacril-scaffold to test their growth-inhibitory potential against wild-type *Fusarium* and phenamacril-resistant strains. Our findings show that most chemical modifications to the phenamacril-scaffold are associated with almost complete loss of fungicidal activity and *in vitro* inhibition of myosin motor domain ATPase activity.

## Introduction

Phenamacril (formerly known by the experimental code JS399-19) belongs to a novel group of single-site acting cyanoacrylate fungicides, which exerts its growth-inhibitory effect against certain members of the much-dreaded genus *Fusarium* [1–5]. Most notably, it is highly effective against *F. graminearum* [1,6], the primary cause of Fusarium head-blight (FHB), a devastating disease of cereals with profound socio-economic consequences [7,8]. Informally classified as a *Fusarium*-specific and an environmentally benign fungicide [9], phenamacril targets *Fusarium* class I myosin [2,10], a member of the ubiquitous eukaryotic myosin superfamily of molecular motor proteins that couple hydrolysis of ATP to the production of force and movement thereby facilitating such diverse and essential actin-associated processes as vesicle- and particle-transport [11,12], phagocytosis [13], septation [14] and cytokinesis [15]. In *F. graminearum*, the phenamacril-mediated inhibition results in mycelial growth defects and aberration of vesicle-transport [2]. This affects both essential cellular processes and more

**Funding:** Funding was provided by The Danish Ministry of Higher Education and Science (grant no. 4005-00204B). M.H.T. was supported by the Volkswagen Stiftung, Niedersächsisches Vorab, Joint Lower Saxony-Israeli Research Projects (Grant VWZN3012) and D.J.M. by DFG grant MA1081/22.1.

**Competing interests:** The authors have declared that no competing interests exist

specialized tasks such as the regulation of deoxynivalenol (DON) biosynthesis and toxisome formation [12].

At the molecular level, phenamacril inhibits the ATPase activity associated with the generic myosin motor domain of *F.spp.* class I myosins [2,10]. While the nucleotide-binding motif is common to all P-loop nucleotide triphosphatases [16,17], four allosteric sites also constitute known binding sites for allosteric effectors of myosin ATPase activity [18–24]. We have recently shown that phenamacril is an allosteric effector and non-competitive inhibitor of the class I myosin motors from *F. graminearum*, *F. avenaceum*, and *F. solani* myosin-1 [25].

Since the establishment of a baseline sensitivity [1,6], several UV-induced phenamacrylic-resistant laboratory and more recently resistant field strains of *Fusarium* and the underlying individual amino acid mutations associated with the resistant phenotypes have been characterized [2,5,10,26,27]. In *F. graminearum*, the mutations confer either low (A135T, V151M, P204S, I434M, A577T, R580G/H and I581F), moderate (S418R, I424R and A577G) or high (K216R/E, S217P/L and E420K/G/D) levels of resistance [28] and mostly involve amino acid residues that are clustered deep within the actin-binding cleft, in immediate vicinity of an allosteric communication pathway responsible for the coupling of the actin and nucleotide binding sites [19,20,29]. While the qualitative resistance mechanism and the emergence of field-resistant strains [5] may render the use of phenamacril short-lived, novel cyanoacrylate derivatives based on the phenamacril-scaffold promise to provide a useful strategy and rational approach to overcome resistant genotypes and preserve the usefulness of this new class of fungicides and inhibitors of myosin ATPase activity.

In this study, we synthesized a small library of 18 cyanoacrylate compounds based on the phenamacril-scaffold and evaluated their growth-inhibitory potential against *F. avenaceum*, *F. solani*, *F. oxysporum*, *F. verticillioides*, *F. graminearum* PH-1 and nine phenamacril-resistant mutants (K216E/R, S217L/P, S418R, E420K/G/D and A577G), which were generated by introducing point-mutations into the *F. graminearum Myo5* locus, which encodes FgMyo1. We subsequently showed a correlation of *in vivo* growth inhibition with an *in vitro* inhibition of the ATPase activity of the purified class I myosin motor domain construct from *F. graminearum* PH-1, thereby demonstrating that cyanoacrylate-compounds based on the phenamacril-scaffold do *de facto* target *Fusarium* class I myosin.

While our findings demonstrate that most substitutions are associated with a significant loss of fungicidal activity, both subtle and large substitutions can sustain or slightly increase the inhibitory potency. While these structure-activity data provide novel insight into the nature of the binding-pocket, they more importantly suggest that overcoming phenamacril-resistance through iterative chemical synthesis of derivatives based on the phenamacril-scaffold is not straightforward.

## Material and methods

### Species and strains

*Fusarium graminearum* PH-1 (NRRL 31084) and *Fusarium avenaceum* 05001 were acquired from the Agricultural Research Service Culture Collection (ARS Culture Collection, USA) and the IBT culture collection at the Danish Technical University, respectively. *Fusarium solani* f. sp. *pisi* 77-14-4 (FGSC 9596), *F. verticillioides* CS7600 (FGSG_7600) and *F. oxysporum* f. sp. *lycopersici* (FGSC 9935) were retrieved from the Fungal Genetics Stock Center (FGSC, USA) [30].

## Vector construction

A mutagenesis vector (pFgMyo1::hph) for targeted double-homologous recombination in the genomic *Myo5* locus (FGSG_01410) was constructed through the four-fragment USER cloning procedure described in Frandsen et. al (2008) [31]. Briefly, to allow for the incorporation of the hph-marker cassette downstream of the *myosin5* gene, left and right borders were amplified with uracil-containing primers and Phusion U Hot Start DNA Polymerase (Thermo Fisher Scientific, USA). The shuttle-vector pRF-HU2 was linearized with PacI (New England Biolabs, USA) and single-stranded overhangs generated by Nt.BbvCI (New England Biolabs, USA). Following gel-purification with the Qiagen gel-extraction kit (Qiagen, Germany) equimolar quantities of vector and border regions were incubated in an uracil-specific excisions reagent mix (USER-mix) (New England Biolabs, USA). Single- and di-nucleotide mutations were subsequently introduced into purified pFgMyo1::hph by the QuickChange II XL site-directed mutagenesis kit (Agilent, USA) according to manufacturer's recommendations and using the primers listed in **S1 Table** in S1 File. Mutations were subsequently verified by Sanger sequencing (Eurofins Genomics, Germany).

## Agrobacterium tumefaciens mediated transformation

Myosin mutagenesis vectors were electroporated into electrocompetent *Agrobacterium tumefaciens* LBA4404 (Thermo Fisher Scientific, USA) using a quartz cuvette (Bio-Rad, USA) and a Gene Pulser II electroporation chamber (Bio-Rad, USA) operated at 2.5 kV, 25 μF and 200 Ω. Following selection and sub-culturing on Lennox Luria-Broth agar with 20 μg·mL$^{-1}$ Rifampicin and 25 μg·mL$^{-1}$ Kanamycin, positive clones were inoculated into liquid IMAS media (0.18% (v/v) glucose, 0.5% (v/v) glycerol and 40% (v/v) 2.5x salt solution (3.625 g·L$^{-1}$ KH$_2$PO$_4$, 5.125 g·L$^{-1}$ K$_2$HPO$_4$, 0.375 g·L$^{-1}$ NaCl, 1.250 g·L$^{-1}$ MgSO$_4$•7H$_2$O, 0.165 g·L$^{-1}$ CaCl$_2$•2H$_2$O, 0.0062 g·L$^{-1}$ FeSO$_4$•7H$_2$O, (NH$_4$)$_2$SO$_4$), 200 μM Acetosyringone (Sigma-Aldrich, USA) and 40 mM MES) and kept at 28˚C and 150 RPM. At OD$_{600}$ • 0.6, they were mixed 1:1 (v/v) with 1 mL 2.5·10$^6$ *F. graminearum* PH-1 macroconidia and distributed onto 10 IMAS agar plates with sterile 80 mm AGF220 filters. At the appearance of an even layer of mycelia (• 2–3 days), filters were transferred onto defined *Fusarium* agar medium (DFM) (pH 6.0 ± 0.1, 1.25% glucose, 10 mM Asparagin, 2.1 mM MgSO$_4$, 11.2 mM KH$_2$PO$_4$, 7 mM KCl, 40 μg·mL$^{-1}$ Na$_2$B$_4$O$_7$•7H$_2$O, 0.4 mg·mL$^{-1}$ CuSO$_4$•7H$_2$O, 1.2 mg·mL$^{-1}$ FeSO$_4$•7H$_2$O, 0.7 mg·mL$^{-1}$ MnSO$_4$•1H$_2$O, 0.8 mg·mL$^{-1}$ NaMoO$_2$•2H$_2$O and 10 mg·mL$^{-1}$ ZnSO$_4$•7H$_2$O) with 150 μg·mL$^{-1}$ Hygromycin B (Invivogen, USA), 0.3 mg·mL$^{-1}$ Cefoxitin (Sigma-Aldrich, USA) with or without 10 μM Phenamacril (Aalborg University, DK). After 3–5 days of incubation at 25˚C, filters were transferred onto fresh DFM plates with 150 μg·mL$^{-1}$ hygromycin B and 10 μM Phenamacril, from which single-colony progeny were obtained.

## Validation of chromosomal integration

Single chromosomal integration of the myosin mutagenesis-cassettes into the *Myo5* locus was verified by long-read whole-genome sequencing (Oxford Nanopore Technology, Oxford, UK). Briefly, the library was prepared from ~1000–2000 ng high-molecular weight (HMW) DNA according to the Ligation Sequencing Kit (SQK-LSK108 or SQK-LSK109 for the Minion and Promethion, respectively) and the Native Barcoding Kit (EXP-NBD103) (Oxford Nanopore Technologies, UK). The libraries were loaded onto primed Flo-min6 (R9.4 chemistry) or FLO-PRO002 (R9.4.1 chemistry) flow-cells and sequenced on the MinION or Promethion (Alfa/Beta) DNA sequencers (Oxford Nanopore Technologies, UK). Reads were base-called by Albacore v.2.0.1 and then trimmed and demultiplexed in Porechop v.0.2.3. The incorporation of the individual mutations was further verified by Sanger Sequencing (Eurofins Genomics,

Germany) of PCR products spanning the relevant codons. All primers are listed in **S1 Table** in
S1 File.

## NADH-coupled steady-state ATPase assay

FgMyo1 (residues 1–711) has been heterologously produced and purified in the baculovirus/
*Sf9* insect cell system (Thermo Fisher Scientific, USA) [25]. ATPase assays were conducted in
accordance with this study and that of Furch et al. (1998) [32]. The actin-activated ATPase
rates were normalized to that of the controls, which had been amended with either 0.5% etha-
nol or 0.5% DMSO.

## Synthesis of phenamacril-derivatives

Synthesis of phenamacril and phenamacril-derivatives are thoroughly described in **S1 Meth-
ods** in S1 File. NMR data were recorded by dissolving each compound in $CDCl_3$ (550 μL),
with the only exception being compound **9**, which for structure validation was dissolved in
DMSO-d6 (500 μL). Reference spectra of Phenamacril, **1**, was also recorded in DMSO-d6. For
all compounds a complete set of $^1H$, $^{13}C$, HSQC and HMBC NMR spectra was measured at
298 K for samples in $CDCl_3$ and 308 K for samples in DMSO-d6. All spectra were recorded on
a BRUKER AVIII-600 MHz NMR spectrometer equipped with a 5 mm CPP-TCI probe.

## Cyanoacrylate stocks

All derivatives were dissolved in either 96% ethanol (VWR Chemicals, DK) or DMSO (Sigma-
Aldrich, USA) as 20 mM stocks.

## Amended agar assays

All amended agar assays were conducted on YPG agar media pH 6.5 ± 0.1 (10 $g·L^{-1}$ yeast
extract, 20 $g·L^{-1}$ peptone, 2.5% (w/v) D-(+)-glucose and 20 $g·L^{-1}$ agar) according to Wollenberg
et. al. [4]. Briefly, D-(+)-glucose, ethanol, phenamacril and phenamacril-derivatives were
added aseptically to autoclaved, cooled media and poured into 5.5 cm vented petri-dishes.
These were inoculated with mycelia plugs (5 mm) from actively growing cultures. The inocu-
lated plates were grown in dark as three replicates at 25 ˚C in an INCU-Line incubator (VWR,
Denmark) with natural convection. Growth was monitored until the controls reached the edge
of the plates (2–5 days depending on the species).

## In silico modeling

The crystal structure of D. discoideum myosin-1E (PDB ID 1LKX) [33] was used as template
for homology modelling of the F. graminearum class I myosin motor domains in the pre-
powerstroke state. The C chain of 1LKX with ADP-vanadate was energy-minimized and
refined in YASARA Structure and WHAT IF ver. 17.8.15 (Yasara2 forcefield, 25 ˚C, TIP3P
water model, 1000 ps simulation, 40 snapshots) [34]. The lowest-energy conformer was subse-
quently used as a template in SWISS-MODEL [35]. Following re-positioning of the ADP-vana-
date, the model was refined using the above parameters.

## Statistical analysis

All plate and ATPase assays were done in six replicates. Error bars denote mean ± S.D. Stu-
dent's two-tailed *t* test was used to compare sample means, with statistical significance
$p \leq 0.05$.

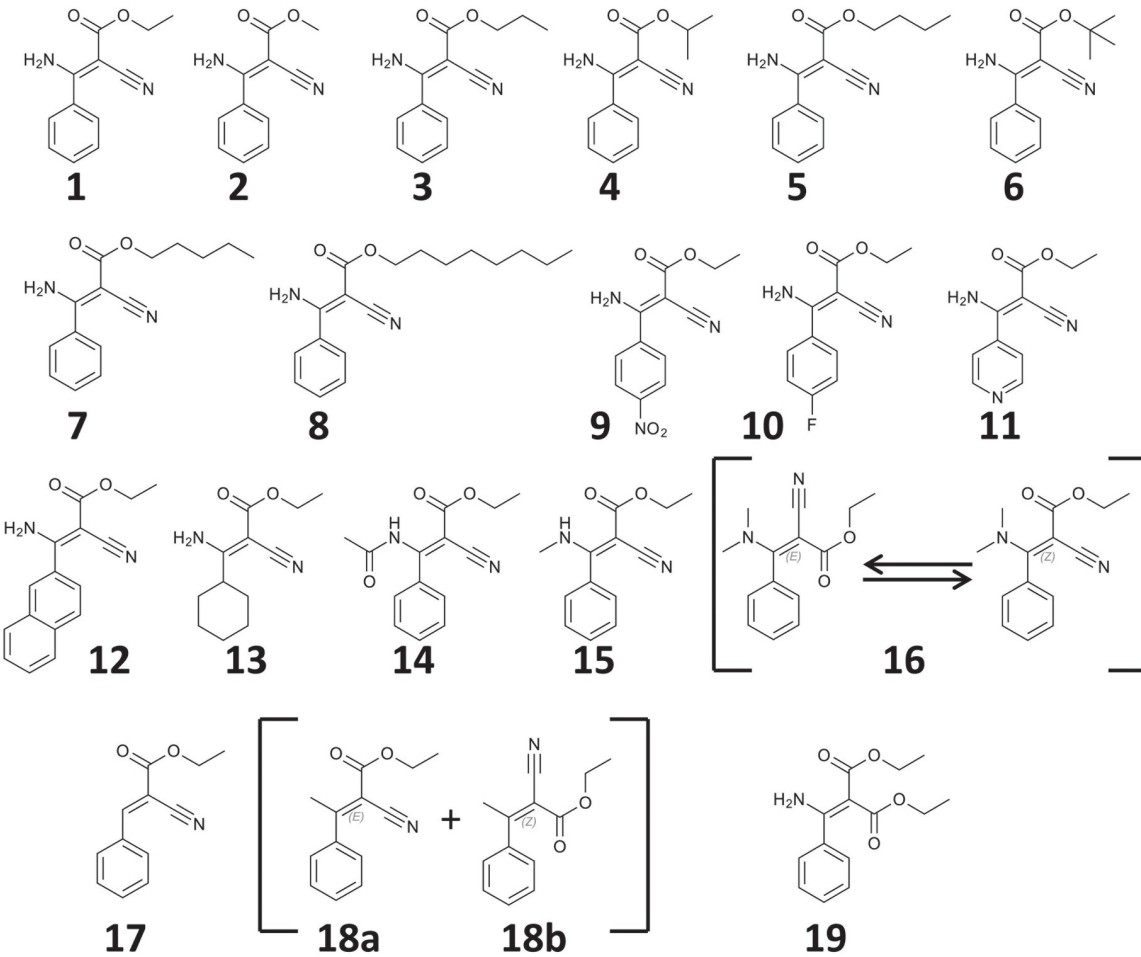

**Fig 1. Chemical structures of phenamacril (1) and its derivatives synthesized.** Compound **16** is in rapid z/e equilibrium and occurs at 47.6% z and 52.4% e. Compounds **18a** and **18b** are not exchanging rapidly, but the mixture obtained after synthesis and purification was used as obtained and contained 69% 18a and 31% **18b**.

## Results and discussion

We synthesized a total of 18 derivatives based on the phenamacril-scaffold (Fig 1). Seven of those (**2–8**) vary only by the alcohol group esterified to the carboxylate. Five compounds (**9–13**) are modified on the aromatic system, five more (**14–18**) feature changes to the amino group, while compound **19** has its cyano-group replaced by another ethyl ester.

It is difficult to distinguish between fungicidal and fungistatic activity working with fungicides. In the case Phenamacril, we have previous demonstrated that Phenamacril predominately works in a fungistatic way in concentrations lower than 100μM inhibiting elongation of hyphae [4].

Our results show that fungicidal activity depends critically on the alcohol moiety esterified to the carboxylic acid. (Fig 2 and **S2 Fig** in S1 File). Specifically, while shortening the alcohol (**2**) resulted in reduction in the ability to inhibit growth of wild-type *F. graminearum*, linearly extending the aliphatic chain (derivatives **3, 5**) either slightly increased or retained most of the fungicidal activity. Longer aliphatic extensions (**7** and **8**) or branching (**4, 6**) of the aliphatic chain reduced the fungicidal potency. These observations collectively suggest that the aliphatic moiety is involved in essential and highly specific interactions of importance to the

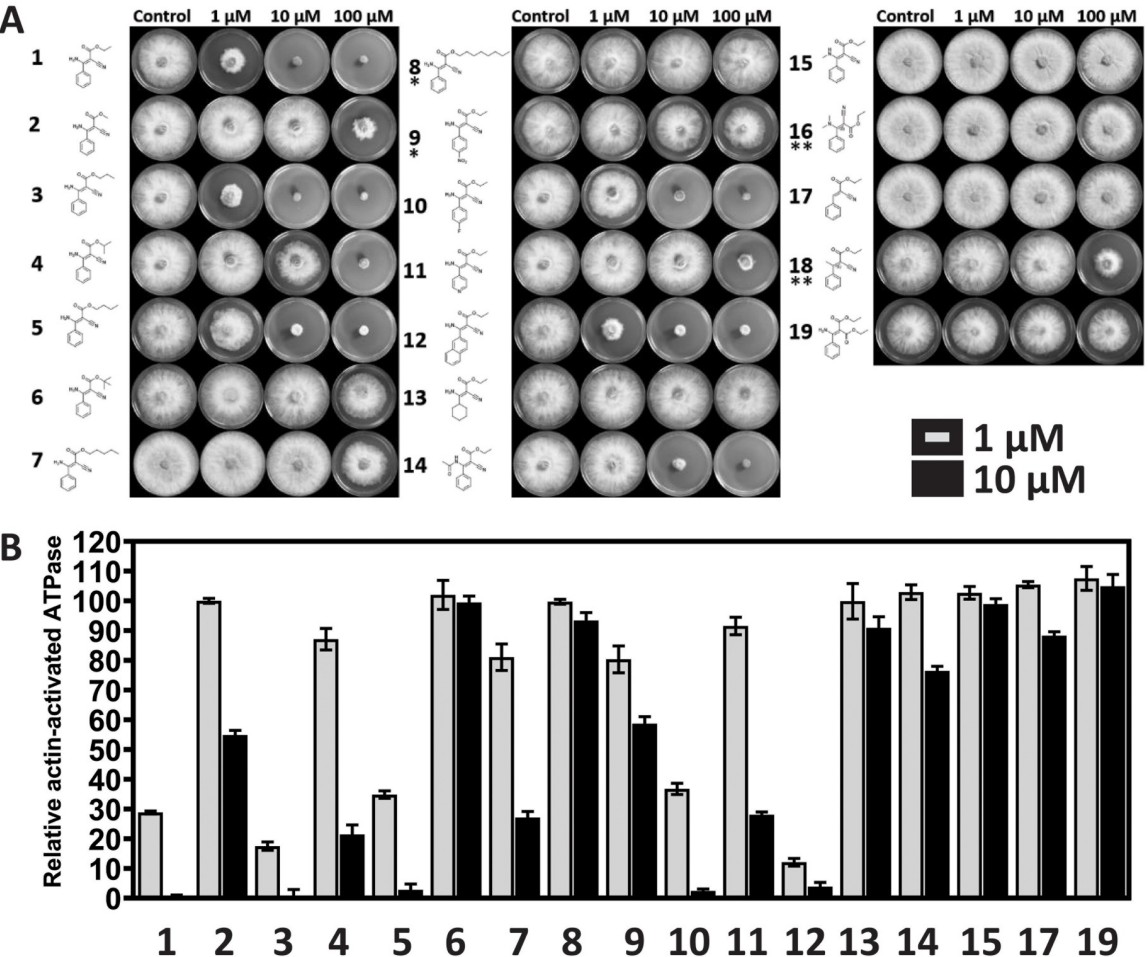

**Fig 2. (A)** The growth-inhibitory effect of phenamacril (**1**) and its derivatives (**2**–**18**) on *F. graminearum* PH-1. Plates were amended with 0–100 μM. Controls were amended with 0.5% ethanol (**1**–**7**, **10**–**19**) or 0.5% DMSO (**8** and **9**). * Derivatives **8** and **9** were assayed at 0, 1, 10 and 20 μM due to low solubility. ** both (*E*) and (*Z*)-configurations are present. **(B)** Normalized actin-activated steady-state ATPase activity of FgMyo1. All values are normalized to the respective 0.5% ethanol (compounds **1**–**7**, **10**–**19**) or DMSO (**8** and **9**) controls. Error bars represent the standard deviations around the mean (n = 6).

mechano-chemical mode of inhibition and that it interacts with a part of the binding-pocket unable to accommodate excessively extended or bulky (aliphatic) substituents. The effect of aromatic substitutions varied significantly. While *p*-Fluor substitution (**10**) sustained the fungicidal activity, introducing an electron-withdrawing group such as in the *p*-nitro derivative (**9**) or substituting the phenyl-moiety for a pyridine-ring (**11**) resulted in significant loss of fungicidal activity. Derivatives **12** and **13** further serve to illustrate that aromaticity (planarity, hydrophobicity and delocalized electro-distribution) is vital to the fungicidal activity. Compared to phenamacril, the non-potent cyclohexyl-derivative has a non-planar ring system and no π-electrons. Conversely, the large, planar, naphthyl-derivative (**12**) maintains an electronic and stereochemical configuration similar to that of phenamacril. It has a growth-inhibitory effect against *F. graminearum*, which indicates that the aromatic-moiety occupies a more exposed part of the binding pocket, in which such a relatively large substitution can be accommodated.

N-methylation (**15**) and *N,N*-di-methylation (**16**) abolish the fungicidal effect but surprisingly the *N*-acetylated derivative (**14**) retains most of the growth-inhibitor effect. Donau et al.

(2017) recently showed that despite an overrepresentation of phenamacril in either the (*E*)-configuration in the newer literature [1,3,6,9,10,36,37] or as a mixture of both the (*E*)- and (*Z*)-configuration [5], phenamacril only exists in its (*Z*)-configuration [38]. In the absence of the intramolecular hydrogen-bond, the electronic structure of phenamacril allows for rotation around the central "double-bond". For derivatives **16** and **18**, this resulted in the presence of both the (*E*)- and (*Z*)-configuration, thus making interpretation difficult. For the *N*,*N*-dimethylated derivative (**16**), the (*Z*)-isomer likely has an effect similar to the *N*-methylated version (**15**). The deaminated derivative (**17**) is similarly ineffective in inhibiting the growth of *F. graminearum*, which is in line with the proposed importance of the hydrogen-bond mediated intra-molecular stabilization. Derivative **18** was the only compound in which the nitrile-group of phenamacril was substituted. Replacing the nitrile group with another ethyl-ester group resulted in a complete loss of fungicidal potency. Whether this is caused by the removal of the nitrile-group, which might act as a hydrogen-bond acceptor, or is the result of introducing a bulkier substituent (*cf.* derivative **4** and **6**) remains to be resolved. However, Wollenberg et al. [25] recently suggested that the nitrile-group occupies a pivotal tight-binding position in the proposed binding pocket, from where it also facilitates inter-molecular hydrogen-bond interactions.

The halogenated pseudilins have demonstrated that subtle chemical modifications to a myosin inhibitor can alter the myosin-class specificity of the inhibitor [19,20,39]. We therefore measured the steady-state ATPase activity of the class I myosin motor domain from *F. graminearum* (FgMyo1) in the presence 0–10 μM cyanoacrylate. Our results showed a general correlation between *in vitro* and *in vivo* inhibition, overall demonstrating that inhibitors based on the phenamacril-scaffold target FgMyo1 (Fig 2). Compounds (**3**) and (**12**) were slightly more potent than phenamacril (**1**) ($p \leq 0.05$) (Fig 2B). Compared to the results from the amended agar plate assays (Fig 2A), the N-acetylated derivative (**15**) seemed to have less inhibitory effect against FgMyo1. Although speculative, it is possible that this compound also has some inhibitory effect against one of the myosin class II, V and XVII members present in *F. graminearum*.

Once we had tested the growth-inhibiting effect of *F. graminearum*, we then proceeded with the testing of a subset of the derivatives directed against the phenamacril-resistant *F. solani* and other relevant crop-pathogenic *Fusarium* spp. but found that the general observations made for *F. graminearum* are also applicable to other phenamacril-susceptible species such as *F. oxysporum*, *F. avenaceum* and *F. verticilliodes* (Fig 3). Although slightly affected by compounds **3** and **5**, *F. solani* remained resistant to all compounds based on the phenamacril-scaffold. The congruence of the responses of the different *Fusarium* species to the inhibitors suggests an identical mechanism of inhibition.

## Resistant *Fusarium* strains

To assess the potential of our phenamacril-derivatives to bypass phenamacril-resistant genotypes, we used *Agrobacterium tumefaciens* mediated transformation to introduce point-mutations into the *Myo5* locus (FGSG_01410) of the *F. graminearum* PH-1 wild-type genome (**S1 Fig** in S1 File). We chose a subset of moderate- and high-resistance conferring mutations [2,10,26], which included the S217L and S217P mutations observed in phenamacril-resistant field strains of *F. fujikuroi* [5]. Interestingly, except maybe for **3**, none of the derivatives tested had any significant effect on the resistant strains (Fig 4). Mutants K216R and S217L were found to be more susceptible to phenamacril than previously reported [2,10,28], with $EC_{50}$ values below the previously characterized A577G and S418R moderate-resistance conferring-mutations [28]. While the reason for this discrepancy remains elusive, Fig 4 more importantly shows that (our) derivatives based on the Phenamacril-scaffold are not able to tackle the

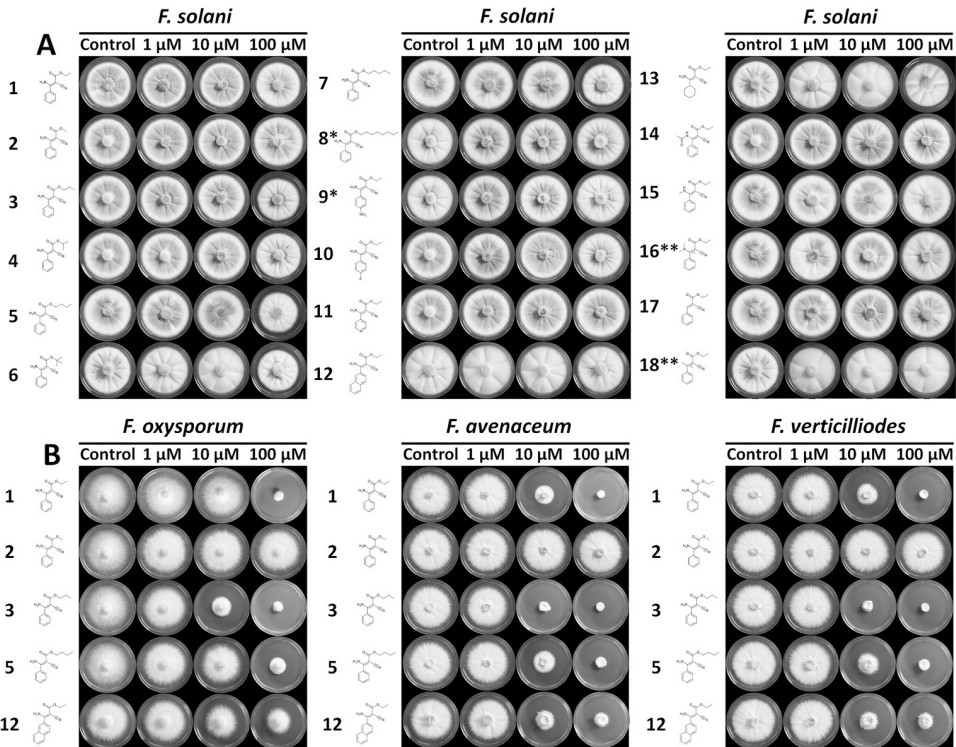

**Fig 3. The effect of a representative subset of cyanoacrylate-derivatives on the growth of the Phenamacril-resistant (A) *F. solani* and (B) Fusarium spp. with differential susceptibility to Phenamacril.** Plates were either amended with 0.5% ethanol or DMSO (control) or 1–100 μM cyanoacrylate. * Derivatives **8** and **9** were assayed at 0, 1, 10 and 20 μM due to low solubility. ** both (*E*) and (*Z*)-configurations are present.

resistant genotypes. The literature mentions some isolated, low-resistance mutations in the motor domain (A135T, V151M and P204S) [28], the presence of which seems to suggest that global or transmitted structural changes can influence the resistance-mechanism. However, they are presumably of less relevance as the recently reported resistant isolates carried the equivalent of the S217L and S217P mutations [5].

Currently, no phenamacril-FgMyo1 structure derived from X-ray diffraction crystallography is available to aid the interpretation of our findings. From one perspective, residues such as K216 in FgMyo1 (one of the amino acids associated with resistance) is known to constitute part of an allosteric communication pathway which transmits information between the nucleotide- and the actin-binding site [19,20,29,39]. These mechano-chemical events involve larger structural changes such as cleft-opening or closure and rotation of the lever arm. This could potentially affect more distal binding-sites. From another perspective, the actin-binding cleft is also known to harbor allosteric binding sites [18–21,24] and the halogenated pseudilins mediate their inhibitory effect through direct interactions with a conserved lysine residue (K216 in *F. graminearum*) [19,20,29,39]. This therefore speaks for a binding of phenamacril in the vicinity of the residues associated with resistance. If this is the case and in light of the loss of effect from subtle modifications to the phenamacril-scaffold (e.g. derivative **2**), it is highly likely that chemical derivatization of phenamacril holds little potential to overcome phenamacril-resistance. This is consistent with our previous findings, in which phenamacril docked in the immediate vicinity of the high- and moderate resistance-conferring amino acid residues in a homology model of FgMyo1 [25]. It is however well-known that proper conclusions require screening of a substantial number of derivatives [40].

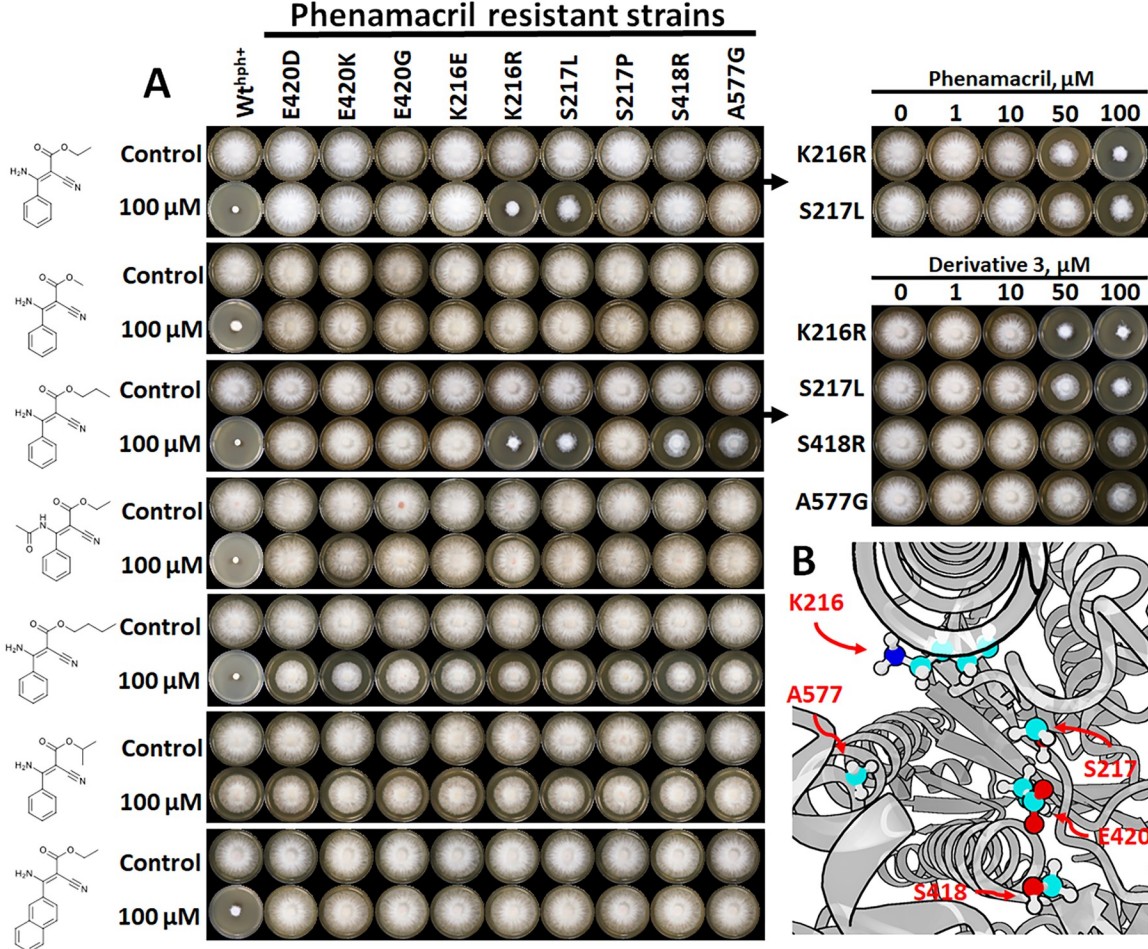

**Fig 4.** (**A**) The effect of phenamacril (**1**) and derivatives **1**, **3**, **4**, **6**, **8** and **10** on the growth of phenamacril-resistant strains of *F. graminearum*. Mutants are named for the resistant-conferring mutation (E420D, E420K, E420G, K216E, K216R, S217L, S418R and A577G). Wt$^{hph++}$ reflects a non-resistant transformant with the hph (Hygromycin B phosphotransferase) cassette inserted into the *Myo5* locus (FGSG_01410). (**B**) Relative position of the resistance-conferring mutations in a homology model of the *F. graminearum* PH-1 class I myosin motor domain.

## Conclusion

We successfully synthesized 18 cyanoacrylate derivatives based on the phenamacril scaffold and while this allowed us to deduce the importance of the different functional groups on phenamacril for the fungicidal potency, we found that only subtle changes could be accommodated without loss of effect. Similarly, none of the derivatives were effective against phenamacril-resistant strains of *Fusarium* to any significant degree, which seems to indicate that derivatization of phenamacril might not be a viable strategy for combatting these emerging resistant genotypes.

## Supporting information

**S1 File.**
(PDF)

## Author Contributions

**Conceptualization:** Rasmus D. Wollenberg, Thorbjørn T. Nielsen, Dietmar J. Manstein, Reinhard Wimmer, Teis E. Sondergaard.

**Data curation:** Rasmus D. Wollenberg, Søren S. Donau, Zoltan Balázs, Sven Giese, Claudia Thiel, Thorbjørn T. Nielsen.

**Formal analysis:** Rasmus D. Wollenberg, Søren S. Donau, Manuel H. Taft, Zoltan Balázs, Claudia Thiel.

**Funding acquisition:** Henriette Giese, Dietmar J. Manstein, Reinhard Wimmer, Teis E. Sondergaard.

**Investigation:** Rasmus D. Wollenberg, Søren S. Donau, Manuel H. Taft, Zoltan Balázs, Sven Giese, Jens L. Sørensen, Teis E. Sondergaard.

**Methodology:** Rasmus D. Wollenberg, Søren S. Donau, Manuel H. Taft, Zoltan Balázs, Jens L. Sørensen, Thorbjørn T. Nielsen, Teis E. Sondergaard.

**Project administration:** Dietmar J. Manstein, Teis E. Sondergaard.

**Validation:** Rasmus D. Wollenberg, Søren S. Donau.

**Writing – original draft:** Rasmus D. Wollenberg, Søren S. Donau.

**Writing – review & editing:** Manuel H. Taft, Henriette Giese, Dietmar J. Manstein, Reinhard Wimmer, Teis E. Sondergaard.

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
