## [Decision Letter · Decision Letter 0]

27 Apr 2020

PONE-D-20-09031

Undefeated – Changing the phenamacril scaffold is not enough to beat resistant Fusarium

PLOS ONE

Dear Prof Søndergaard,

Thank you for submitting your manuscript to PLOS ONE. After careful consideration, we feel that it has merit but does not fully meet PLOS ONE’s publication criteria as it currently stands. Therefore, we invite you to submit a revised version of the manuscript that addresses the points raised during the review process.

The manuscript has been reviewed by two experts in the field; please find their comments appended at the end of this email. Reviewer #2 in particular identified a number of issues. Based on the reviewers’ comments as well as my own reading of the manuscript I have decided for ‘Major Revision’ as some experiments, statistics, and other analyses are not performed at the required high technical standard and not described in sufficient detail, respectively.

Specifically, the following points needs to be addressed in a revised version of this manuscript:

1. The agar assay description in Materials and Methods needs to be improved to include all relevant environmental conditions and experimental parameters.

2. Materials and Methods as well as individual experimental section need to be updated to include appropriate description of the statistical methodologies and analyses used. The description and presentation of results needs to include relevant statistical data and critical appraisal based on those data.

3. In the Supplementary Information, the description of all chemical compounds synthesised needs to include experimental yields and purity data as well as NMR, mass spectrometric (MS) and, if available, HPLC and/or any other physico-chemical characterisation data. NMR spectral data need to be assigned, and MS data need to explained.

4. The manuscript needs to be thoroughly edited such as to provide valid referencing throughout.

5. In addition to the above points, all issues raised by the reviewers need to be addressed.

We would appreciate receiving your revised manuscript by Jun 11 2020 11:59PM. To enhance the reproducibility of your results, we recommend that if applicable you deposit your laboratory protocols in protocols.io, where a protocol can be assigned its own identifier (DOI) such that it can be cited independently in the future. For instructions see: http://journals.plos.org/plosone/s/submission-guidelines#loc-laboratory-protocols

We look forward to receiving your revised manuscript.

Kind regards,

Andreas Hofmann

Academic Editor

PLOS ONE

Journal Requirements:

2. Please ensure that you refer to Figures 1 to 4 in your text as, if accepted, production will need this reference to link the reader to the figure.

Additional Editor Comments (if provided):

Reviewers' comments:

Reviewer's Responses to Questions

**Comments to the Author**

1. Is the manuscript technically sound, and do the data support the conclusions?

Reviewer #1: Yes

Reviewer #2: No

2. Has the statistical analysis been performed appropriately and rigorously? 

Reviewer #1: N/A

Reviewer #2: No

3. Have the authors made all data underlying the findings in their manuscript fully available?

Reviewer #1: Yes

Reviewer #2: No

4. Is the manuscript presented in an intelligible fashion and written in standard English?

Reviewer #1: Yes

Reviewer #2: Yes

5. Review Comments to the Author

Reviewer #1: The manuscript is very well described, with clear objectives and an appropriate methodology. The results of the analysis of the new molecules show us that structural changes do not always cause improvements in biological activity / toxicity. It is very difficult to find a chemical analog with an activity superior to the base compound, however planned and rational the synthesis process may be. Some less adjustments are needed:

- Results and discussion: lines 182/183/231/233, correct: (Error! Reference source not found.).

- Line 197: lots of space.

- Lines 253/256: Missed mentioning the figure number.

- The quality of the chemical structures in figures 3 and 4 is poor.

Reviewer #2: I found the manuscript very intriguing in its hypotheses, and I’m very supportive with researches aimed at demonstrating using indirect strategies or studies. Unfortunately, I have many conceptual concerns about the present manuscript, reported as follows:

Mat&Met: The section about the agar assay lacks of the culture incubation conditions (temperature, light/dark, approximate number of incubation days). It is a key information that must be provided. Additionally, no statistical analysis of growth-inhibition data seems to have been performed, since neither information about analysis software is described nor p values are indicated in charts or their legends, while something suddenly appears in Line 232 (“were slightly more potent than phenamacril (1) (P < 0.05).”)

The chemical part is very wanting. The characterisation of the compounds has been made only by NMR and no reaction yields are reported.

Results & Discussion

In this paragraph there are at least 5 references missing (Error! Reference source not found)

An evaluation of inhibitory effect of compounds was described. Although visual data were shown as pictures of strains cultures, no measurement was conducted to numerically estimate differences in growth rate…thus, I wonder how Authors could discuss and then correlate such differences to the biological activity of compounds (for example in Line 192: “branching of the aliphatic chain reduced the fungicidal potency by ≈100-fold”).

Another consideration I would like to raise is about the “fungicide” potential of compounds: standing to a simple radial growth assay, as reported in the present work, it is not possible to distinguish a fungicidal activity from a fungistatic one. In fact, as frequently demonstrated, inhibitory effect of bioactives on fungi often rely on a strong delay of hyphae elongation, To be able to assess the difference, viability assay and biomass accumulation measurement should be done.

Minor issues:

Line 135: “w/wo” is this abbreviation allowed by the Journal?

Line 182, 188, 231, 233, 241: reference error highlighted.

Line 253, 256: figure number is missing

Figure 4: the relative position of the resistance-conferring mutations in the homology model is shown, but it is not clear if this docking analysis was performed by Author specifically for this work (if this is the case, methodological information in the proper section is missing) or if it is reported for a reader’s better comprehension only.

6. PLOS authors have the option to publish the peer review history of their article (what does this mean?). If published, this will include your full peer review and any attached files.

Reviewer #1: Yes: Daiane Flores Dalla Lana

Reviewer #2: No

---

## [Author Response · Author response to Decision Letter 0]

10 Jun 2020

We would like to thanks the reviewers for the comments.

Reviewer #1: The manuscript is very well described, with clear objectives and an appropriate methodology. The results of the analysis of the new molecules show us that structural changes do not always cause improvements in biological activity / toxicity. It is very difficult to find a chemical analog with an activity superior to the base compound, however planned and rational the synthesis process may be. Some less adjustments are needed:

- Results and discussion: lines 182/183/231/233, correct: (Error! Reference source not found.). Thanks, corrected.

- Line 197: lots of space. Corrected.

- Lines 253/256: Missed mentioning the figure number. Corrected

- The quality of the chemical structures in figures 3 and 4 is poor. We have made new figures 3 and 4 in better quality.

Reviewer #2: I found the manuscript very intriguing in its hypotheses, and I’m very supportive with researches aimed at demonstrating using indirect strategies or studies. Unfortunately, I have many conceptual concerns about the present manuscript, reported as follows:

Mat&Met: The section about the agar assay lacks of the culture incubation conditions (temperature, light/dark, approximate number of incubation days). It is a key information that must be provided. Additionally, no statistical analysis of growth-inhibition data seems to have been performed, since neither information about analysis software is described nor p values are indicated in charts or their legends, while something suddenly appears in Line 232 (“were slightly more potent than phenamacril (1) (P < 0.05).”)

We have added the following: These were inoculated with mycelia plugs (5 mm) from actively growing cultures. The inoculated plates were grown in dark as three replicates at 25°C in an INCU-Line incubator (VWR, Denmark) with natural convection. Growth was monitored until the controls reached the edge of the plates (2-5 days depending on the species).

And added a new paragraph: “Statistical analysis

All plate and ATPase assays were done in triplicates. Error bars denote mean ± S.D. Student's two-tailed t test was used to compare sample means, with statistical significance p > 0.05.”

The chemical part is very wanting. The characterisation of the compounds has been made only by NMR and no reaction yields are reported.

We have added all NMR spectra in supplemental materials and added the yield to the table in supplemental methods. 

Results & Discussion

In this paragraph there are at least 5 references missing (Error! Reference source not found)

Corrected.

An evaluation of inhibitory effect of compounds was described. Although visual data were shown as pictures of strains cultures, no measurement was conducted to numerically estimate differences in growth rate…thus, I wonder how Authors could discuss and then correlate such differences to the biological activity of compounds (for example in Line 192: “branching of the aliphatic chain reduced the fungicidal potency by ≈100-fold”). 

We agree that 100 fold is an unsuccessful term to use. It was meant that the inhibitory effect changed from 1uM to 100 uM. We have canceled the numbers.

Another consideration I would like to raise is about the “fungicide” potential of compounds: standing to a simple radial growth assay, as reported in the present work, it is not possible to distinguish a fungicidal activity from a fungistatic one. In fact, as frequently demonstrated, inhibitory effect of bioactives on fungi often rely on a strong delay of hyphae elongation, To be able to assess the difference, viability assay and biomass accumulation measurement should be done.

We completely agree and we have previous (ref. 4) been concerned about that fact (4. Wollenberg RD, Donau SS, Nielsen TT, Sorensen JL, Giese H, et al. (2016) Real-time imaging of the growth-inhibitory effect of JS399-19 on Fusarium. Pestic Biochem Physiol 134: 24-30. Based on our observation and combined with the literature, we find that Phenamacril possess both fungicidal and fungistatic effects dependent by concentration.

Minor issues:

Line 135: “w/wo” is this abbreviation allowed by the Journal? Changed

Line 182, 188, 231, 233, 241: reference error highlighted. It refers to figures because of automated track not recognized in PlosOnes system. Changed.

Line 253, 256: figure number is missing Changed

Figure 4: the relative position of the resistance-conferring mutations in the homology model is shown, but it is not clear if this docking analysis was performed by Author specifically for this work (if this is the case, methodological information in the proper section is missing) or if it is reported for a reader’s better comprehension only.

Methodological information has been added

In silico modeling

The crystal structure of D. discoideum myosin-1E (PDB ID 1LKX) (Kollmar et al., 2002) was used as template for homology modelling of the F. graminearum class I myosin motor domains in the pre-powerstroke state. The C chain of 1LKX with ADP-vanadate was energy-minimized and refined in YASARA Structure and WHAT IF ver. 17.8.15 (Yasara2 forcefield, 25�C, TIP3P water model, 1000 ps simulation, 40 snapshots) (Krieger and Vriend, 2014). The lowest-energy conformer was subsequently used as a template in SWISS-MODEL (Bordoli et al., 2009). Following re-positioning of the ADP-vanadate, the model was refined using the above parameters. 

Three refs from this section have been added to the literature list.

---

## [Decision Letter · Decision Letter 1]

15 Jun 2020

PONE-D-20-09031R1

Undefeated – Changing the phenamacril scaffold is not enough to beat resistant Fusarium

PLOS ONE

Dear Dr. Søndergaard,

Thank you for submitting your manuscript to PLOS ONE. After careful consideration, we feel that it has merit but does not fully meet PLOS ONE’s publication criteria as it currently stands. Therefore, we invite you to submit a revised version of the manuscript that addresses the points raised during the review process.

The revised version of this manuscript has been sent to both reviewers of the original submission and reviewer #1 is satisfied with the present version. However, based on my own reading of the mansucript, I have decided for 'Major Revision', as one issue raised in the previous reviews has not been considred in the revised version of the manuscript. Additionally, there are inconsistencies regarding the statitical analyses which need to be addressed.

A revised version of this manuscript should address the following points:

1. Statistical analyses: Line 190 says n = 3, but in line 427 it says n = 6.

2. Fungicidal vs. fungistatic activity: I appreicate your response to the issue raised by reviewer #2. However, I can't see that this has point has been taken into account in the revised manuscript. If you believe that the reviewer's comment can be addressed by previous results/studies, then this needs to be discussed in the main text with appropriate literature being referenced.

3. Line 191: "Student's two-tailed t test was used to compare sample means, with statistical significance p > 0.05."

Presumably, this should be p <= 0.05.

4. Line 424: "Derivatives 8 and 9 were due low solubility assayed at 0, 1, 10 and 20 μM."

Please amend to: Derivatives 8 and 9 were assayed at 0, 1, 10 and 20 μM due low solubility.

5. Supplementary methods, compounds table: yield should be given with only one significant decimal after the floating point or as integer values.

6. Throughout (manuscript and supplemental information):

a. Please use consistent upper or, preferrably, lower case 'p' to denote the p-value.

b. Consistently add space between the number and '°C' when giving temperatures(e.g. '0 °C' instead of '0°C').

c. check spelling, in particular the name of chemicals (e.g. dichloromethane instead of Dichloromethane).

d. Variables should be set in italics, e.g. <i>t</i>-test, <i>p</i>-value, etc.

We look forward to receiving your revised manuscript.

Kind regards,

Andreas Hofmann

Academic Editor

PLOS ONE

Reviewers' comments:

Reviewer's Responses to Questions

**Comments to the Author**

1. If the authors have adequately addressed your comments raised in a previous round of review and you feel that this manuscript is now acceptable for publication, you may indicate that here to bypass the “Comments to the Author” section, enter your conflict of interest statement in the “Confidential to Editor” section, and submit your "Accept" recommendation.

Reviewer #1: All comments have been addressed

2. Is the manuscript technically sound, and do the data support the conclusions?

Reviewer #1: Yes

3. Has the statistical analysis been performed appropriately and rigorously? 

Reviewer #1: N/A

4. Have the authors made all data underlying the findings in their manuscript fully available?

Reviewer #1: Yes

5. Is the manuscript presented in an intelligible fashion and written in standard English?

Reviewer #1: Yes

6. Review Comments to the Author

Reviewer #1: (No Response)

7. PLOS authors have the option to publish the peer review history of their article (what does this mean?). If published, this will include your full peer review and any attached files.

Reviewer #1: No

---

## [Author Response · Author response to Decision Letter 1]

16 Jun 2020

A revised version of this manuscript should address the following points:

1. Statistical analyses: Line 190 says n = 3, but in line 427 it says n = 6. Corrected, it was six replicates

2. Fungicidal vs. fungistatic activity: I appreicate your response to the issue raised by reviewer #2. However, I can't see that this has point has been taken into account in the revised manuscript. If you believe that the reviewer's comment can be addressed by previous results/studies, then this needs to be discussed in the main text with appropriate literature being referenced.

We have added the following to results and discussion: 

It is difficult to distinguish between fungicidal and fungistatic activity working with fungicides. In the case Phenamacril, we have previous demonstrated that Phenamacril predominately works in a fungistatic way in concentrations lower than 100µM inhibiting elongation of hyphae [4]. 

3. Line 191: "Student's two-tailed t test was used to compare sample means, with statistical significance p > 0.05."

Presumably, this should be p <= 0.05. Corrected

4. Line 424: "Derivatives 8 and 9 were due low solubility assayed at 0, 1, 10 and 20 μM."

Please amend to: Derivatives 8 and 9 were assayed at 0, 1, 10 and 20 μM due low solubility. Changed

5. Supplementary methods, compounds table: yield should be given with only one significant decimal after the floating point or as integer values. Corrected

6. Throughout (manuscript and supplemental information):

a. Please use consistent upper or, preferrably, lower case 'p' to denote the p-value. Corrected

b. Consistently add space between the number and '°C' when giving temperatures(e.g. '0 °C' instead of '0°C'). Corrected

c. check spelling, in particular the name of chemicals (e.g. dichloromethane instead of Dichloromethane). Corrected

d. Variables should be set in italics, e.g. *t*-test, *p*-value, etc. Corrected

---

## [Editor Report · Decision Letter 2]

18 Jun 2020

Undefeated – Changing the phenamacril scaffold is not enough to beat resistant Fusarium

PONE-D-20-09031R2

Dear Dr. Søndergaard,

We’re pleased to inform you that your manuscript has been judged scientifically suitable for publication and will be formally accepted for publication once it meets all outstanding technical requirements.

Kind regards,

Andreas Hofmann

Academic Editor

PLOS ONE
---

## [Editor Report · Acceptance letter]

19 Jun 2020

PONE-D-20-09031R2 

Undefeated – Changing the phenamacril scaffold is not enough to beat resistant *Fusarium*

Dear Dr. Sondergaard:

I'm pleased to inform you that your manuscript has been deemed suitable for publication in PLOS ONE. Congratulations! Your manuscript is now with our production department. 

Kind regards, 

on behalf of

Associate Professor Andreas Hofmann 

Academic Editor

PLOS ONE